# Evaluation of Aciclovir-Induced Nephrotoxicity in Critically Ill Patients: A Propensity-Matched Cohort Study

**DOI:** 10.3390/jcm14051409

**Published:** 2025-02-20

**Authors:** René J. Boosman, Rob J. Bosman, Peter H. J. van der Voort, Eric J. F. Franssen

**Affiliations:** 1Department of Clinical Pharmacy, OLVG Hospital, 1091 AC Amsterdam, The Netherlands; 2Department of Intensive Care, OLVG Hospital, 1091 AC Amsterdam, The Netherlands; 3TIAS School for Business and Society, Tilburg University, 5037 AB Tilburg, The Netherlands

**Keywords:** nephrotoxicity, aciclovir, critically ill patients, ICU, propensity score matching

## Abstract

**Background/Objectives:** Aciclovir is a widely used antiviral agent. Since aciclovir is primarily eliminated through the kidneys, maintaining renal function is crucial to avoid toxicity. Although mitigating strategies are introduced in the standard of care, nephrotoxicity is still a major concern during treatment, especially for critically ill intensive care unit (ICU) patients. Therefore, risk factors for the development of nephrotoxicity during aciclovir therapy should be addressed. This study aimed to evaluate if aciclovir in combination with therapeutic drug monitoring (TDM) and additional nephrotoxicity-mitigating strategies is associated with a decrease in renal function in critically ill ICU patients. **Methods:** In a cohort of ICU patients with or without intravenous aciclovir treatment (including standard of care mitigating strategies) propensity score matching was applied to balance baseline characteristics between aciclovir-treated and untreated groups. Aciclovir was monitored by measuring serum levels and the dose was adjusted when needed. Renal function was primarily assessed through serum creatinine. Univariate and multivariate regression analyses were used to identify risk factors for nephrotoxicity during ICU admission. **Results**: After propensity score matching, the study included 518 ICU patients, of whom 259 received aciclovir. Aciclovir was not associated with a significant decrease in renal function during admission. In fact, renal function appeared to improve in the aciclovir-treated group compared to the control group (beta-coefficient: −14.5 (95% confidence interval: −28.3 to −0.68), *p = 0.04*). Median aciclovir concentrations remained within the exploratory therapeutic range. **Conclusions**: Aciclovir therapy, at least when appropriately monitored, does not independently induce nephrotoxicity in critically ill ICU patients. TDM may further enhance safety by preventing supratherapeutic drug exposures. The results are significant as they provide evidence supporting the safe use of aciclovir in a vulnerable patient population. Future studies should focus on establishing therapeutic and toxic concentration thresholds for aciclovir and assessing the clinical utility of TDM in this context.

## 1. Introduction

Herpes simplex virus (HSV) type 1 or type 2 seropositivity is common in the global population with a prevalence up to 80–85% [1]. In critically ill patients, especially those in the intensive care unit (ICU), viral reactivation (with e.g., HSV or varicella zoster virus (VZV)) is common and can lead to severe complications and mortality [2,3]. Aciclovir, a nucleoside analogue, is widely used to treat and prevent the reactivation of HSV and VZV in a standard intravenous dosing regimen of 5–10 mg/kg administered three times daily [4].

Aciclovir is predominantly eliminated via the kidneys, as approximately 75–80% is recovered in the urine as the unchanged drug in the first 24 h after administration [4]. Moreover, in patients with severe renal dysfunction (estimated glomerular filtration rate (eGFR): 29 mL/min/1.73 m^2^), the total clearance of aciclovir is decimated when compared with the clearance in patients with adequate renal function [5]. Therefore, renal impairment can result in high exposure to aciclovir. Higher aciclovir exposure might also result in more toxicity since it has been suggested that a lower renal function (eGFR < 45 mL/min) is a predictor for the development of adverse events during aciclovir treatment [6,7]. Consequently, the drug label warrants dose adjustments for patients with renal function below 50 mL/min to avoid additional toxicity [4].

Avoiding drug toxicity is especially important for critically ill patients in the ICU. ICU patients are at an increased risk of organ toxicities, including nephrotoxicity, due to the severity of illness, underlying chronic conditions, drug–drug interactions, and altered pharmacokinetics [8]. Moreover, aciclovir itself has a nephrotoxic potential [9]. Previous studies reported renal impairment in up to 48% of patients receiving high doses of aciclovir [10,11]. This nephrotoxicity primarily results from intratubular crystallization of the drug, leading to obstructive nephropathy and acute kidney injury (AKI) [9,12]. Besides nephrotoxicity, accumulation of aciclovir can also result in neurological, gastrointestinal, and hematological toxicity [4], highlighting the importance to prevent excessive drug exposure.

Current strategies to mitigate the risk of aciclovir-induced nephrotoxicity include dose adjustments based on renal function, hyperhydration, and avoiding high-dose administration [13]. While therapeutic drug monitoring (TDM) has been suggested to individualize aciclovir dosing in ICU patients, there is limited knowledge about the relationship between drug exposure, efficacy, and toxicity [14]. Various risk factors, such as age, days of treatment and use of nephrotoxic medication other than aciclovir, for the development of nephrotoxicity during aciclovir treatment are identified. However, these studies are often performed outside the ICU population [15,16,17].

This study aims to investigate the association between aciclovir treatment, including standard care elements such as TDM, and the development of renal impairment in ICU patients. By exploring this relationship, we seek to identify risk factors that predispose ICU patients to aciclovir-induced nephrotoxicity, thereby providing insights to optimize dosing strategies that balance therapeutic efficacy with the risk of nephrotoxicity in critically ill patients.

## 2. Materials and Methods

### 2.1. Study Design and Patients

This retrospective, propensity-matched, observational cohort study was conducted in the ICU at OLVG hospital in Amsterdam, the Netherlands.

The study included all patients admitted to the ICU, OLVG location east, between January 2008 and November 2023. The study population was divided into two cohorts based on whether patients received intravenous aciclovir therapy. Patients were excluded when they were admitted to the ICU for less than 72 h or when they received renal replacement therapy (RRT) at inclusion. Additionally, in the aciclovir group, patients were further excluded if they received aciclovir therapy for less than 24 h.

### 2.2. Data Collection

Data were extracted from the ICU patient data management system MetaVision^®^ (iMDsoft, Tel Aviv, Israel). Patient characteristics, including demographics (age, biological gender, body mass index (BMI)), comorbidities on inclusion (immunological insufficiency, diabetes, chronic kidney disease (CKD), and AKI), potential nephrotoxic co-medication (use of NSAIDs, vancomycin, or aminoglycosides), and laboratory renal markers (serum creatinine, serum urea, diuresis), were collected. The estimated glomerular filtration rate (eGFR) was calculated based on the Chronic Kidney Disease Epidemiology Collaboration (CKD-EPI) [18]. Severity scores (APACHE III score, APACHE IV predicted mortality, sequential organ failure assessment (SOFA), Renal SOFA, and Risk, Injury, Failure, Loss, End stage renal disease (RIFLE score)) and information on the mechanical ventilation status and RRT were defined according to the Dutch National Intensive Care Registry (NICE; https://www.stichting-nice.nl, accessed on 5 January 2025). For aciclovir-treated patients, data on therapy duration, cumulative aciclovir dose, and aciclovir trough concentrations were also collected.

In the ICU, aciclovir is administered as a 60 min pump-driven intravenous infusion via a central venous catheter at a dose of 5–10 mg/kg three times daily. Per internal protocol, aciclovir trough levels were measured after at least 24 h of therapy and subsequently every other day. Serum levels were determined in the laboratory of the hospital pharmacy using a validated ultra-high-performance liquid chromatography method coupled with mass spectrometry (UHPLC-MS/MS; TSQ Quantum Acces Max, Thermo Scientific, Waltham, MA, USA). Briefly, 50 µL of serum sample was treated with 500 µL of the stable isotope internal standard aciclovir-D4 (0.3 µg/mL) in acetonitrile. Calibration curves were constructed by adding known amounts of aciclovir to blank human serum (range 0.25–5 mg/L). All samples were injected in positive ionization mode of the UHPLC-MS/MS containing a hydrophilic interaction chromatography column (HILIC 3 µm, 2.1 × 100 mm including a Vanguard column (Waters Co, Milford, MA, USA)), and an isocratic eluent containing ammoniumformate (9 mM), formic acid (0.1%), and acetonitrile was used with a flow of 0.4 mL/min. Aciclovir and D4-aciclovir were quantified using the transition of mass/charge (*m*/*z*) of 226.0 → 152.2 *m*/*z* and 230.1 → 152.0 *m*/*z*, respectively. Reference trough levels were set at 0.5–3 mg/L [4], primarily to prevent neurotoxicity and nephrotoxicity. Concentrations above 4.5 mg/L were considered toxic [19]. Trough concentrations were interpreted by a hospital pharmacist, and non-trough levels were excluded from analysis.

### 2.3. Endpoints

The primary endpoint was the difference between the mean serum creatinine concentration during aciclovir therapy (for the aciclovir group) or during ICU admission (for the no-aciclovir group) and the baseline serum creatinine concentration. Creatinine, rather than eGFR, was chosen as the primary endpoint due to its robustness, aligning with methods used in prior nephrotoxicity studies involving ICU patients, ensuring consistency with established research practices. Baseline was defined as the serum creatinine value at the start of aciclovir therapy (within 24 h) for the aciclovir group or at the time of ICU admission for the no-aciclovir group. Secondary endpoints included differences in mean serum urea, SOFA score, RIFLE score, and the eGFR between baseline and during therapy or admission. Additionally, the number of patients developing AKI (as defined by Kidney Disease Improving Global Outcomes (KDIGO) [20]) or requiring RRT during aciclovir therapy (for the aciclovir group) or during ICU admission (for the no-aciclovir group) was calculated.

### 2.4. Statistical Analysis

Statistical differences between patient characteristics in both groups were assessed using chi-squared tests, Fisher exact tests, or *t*-tests, as appropriate and based upon normality testing using the Shapiro–Wilk test. A *p*-value < 0.05 was considered statistically significant, whereas a Bonferroni correction was applied to correct for testing on multiple covariates. Statistical analyses were performed using R (version 4.4.1, R-project, Vienna, Austria).

An unmatched analysis was conducted on the entire cohort meeting inclusion and exclusion criteria. To correct for confounding in the heterogeneous population of the ICU a propensity score matching was used. Propensity score matching (nearest neighbor) was performed on age, APACHE III score, baseline creatinine, baseline SOFA, baseline RIFLE, length of stay in the ICU (LOS) and number of nephrotoxic medications, with a 1:1 matching ratio and a tolerance score of 0.05.

Univariate and multivariate Cox regression analyses were used to assess the effect of individual covariates on the primary outcome. Logically relevant covariates were included in the multivariate analysis. These risk factors included aciclovir therapy as well as the cumulative dose of aciclovir, age, obesity (BMI ≥ 30 kg/m^2^ [17]), LOS, and concurrent use of nephrotoxic drugs. Moreover, the effect of baseline renal markers (creatinine, urea, RIFLE score and SOFA score, and RRT) on the primary endpoint was assessed. To assess adequate matching the propensity score was also considered as a covariate in the multivariate analysis.

## 3. Results

During the study period, 7770 eligible patients were identified. Among them, 5277 patients were included in the analysis, with 259 patients treated with aciclovir (aciclovir group) and 5018 not receiving aciclovir (no-aciclovir group). Figure 1 illustrates the flow chart of the selected patients including those who were excluded in the final analysis.

Baseline characteristics for patients in the unmatched and (propensity score) matched cohorts are provided in Table 1. Moreover, in Appendix A the changes in absolute standardized differences of the included covariates before and after propensity score matching are shown. Missing data in the unmatched cohort included the APACHE IV probability (n = 22, all in the no-aciclovir group), SOFA score at baseline (n = 1, in the no-aciclovir group), and the urea concentration at baseline (n = 715 in the no-aciclovir group and n = 1 in the aciclovir group). After matching, missing data were only found in the urea concentration at baseline (n = 19 for the no-aciclovir group and n = 1 for the aciclovir group). Bonferroni correction led to a significance level of 0.0023. Overall, the baseline demographics were comparable between the aciclovir and no-aciclovir groups. However, in the unmatched cohort, the APACHE III score (80 vs. 76, *p* < 0.001) and APACHE IV predicted mortality (0.35 vs. 0.27, *p* < 0.001) were higher in the aciclovir group compared to the no-aciclovir group. Patients in the aciclovir group were also more likely to have immunological insufficiency at baseline (25% vs. 10%, *p* < 0.001) and had statistically significantly higher urea concentrations at baseline (16 mmol/L vs. 9.9 mmol/L, *p* < 0.001) when compared to the no-aciclovir group. Additionally, the aciclovir-treated patients were less frequently exposed to nephrotoxic co-medications (80.3% vs. 64.1% received no nephrotoxic co-medication, *p* < 0.001) and had longer LOS (16.5 days vs. 5.9 days, *p* < 0.001).

After matching for relevant variables, differences between the two groups were reduced, but some remained statistically significant. Specifically, LOS (16.5 vs. 11.1 days, *p* < 0.001) and baseline urea concentrations (16.0 mmol/L vs. 9.7 mmol/L, *p* < 0.001) were still notably different between the aciclovir and non-aciclovir groups.

A total of 787 aciclovir samples for 254 (98%) patients were collected during the inclusion period, with a median of 2 samples per patient (range: 1–15 samples). Six samples were excluded from this analysis because these samples were not drawn at a trough level. A population median trough concentration of 1.59 mg/L (0.87–3.25 mg/L) was found.

### 3.1. Univariate Analysis

#### 3.1.1. Primary Outcome

The results of the univariate analyses are shown in Table 2. In the unmatched cohort, the delta creatinine levels were lower in the aciclovir group compared to the no-aciclovir group (−10.5 (−34.1 to 2.50) vs. −4.4 (−19.7 to 7.40), *p* = 0.09), though the difference did not reach statistical significance. Similarly, in the matched cohort, there was no significant difference in delta creatinine between the aciclovir and no-aciclovir groups (−10.5 (−34.1 to 2.5) vs. −5.5 (−22.4 to 3.1), *p* = 0.22).

#### 3.1.2. Secondary Outcomes

Delta urea levels showed a significant decrease in the aciclovir group compared to an increase in the no-aciclovir group in both the unmatched (−0.49 (−3.9 to 3.5) vs. 2.0 (−0.4 to 4.8), *p* < 0.001) and matched cohorts (−0.49 (−3.9 to 3.5) vs. 1.9 (−0.5 to 5.4), *p* < 0.001). For delta SOFA scores, a greater reduction was observed in the no-aciclovir group compared to the aciclovir group in both the unmatched (−1.9 (−3.2 to 0.8) vs. −1.3 (−3.0 to 0.1), *p* < 0.001) and matched cohorts (−2.0 (−3.5 to −0.7) vs. −1.3 (−3.0 to 0.1), *p* < 0.001). Delta RIFLE scores showed no significant difference between the groups in both the unmatched (*p* = 0.71) and matched cohorts (*p* = 0.44). Delta eGFR was significantly higher in the aciclovir group compared to the no-aciclovir group in the unmatched cohort (6.3 (−1.3 to 17.5) vs. 2.5 (−3.9 to 9.7), *p* < 0.001), but this difference was not significant in the matched cohort (*p* = 0.23).

The incidence of acute kidney injury (AKI) was lower in the aciclovir group compared to the no-aciclovir group in both the unmatched (1.9% vs. 4.4%, *p* = 0.08) and matched cohorts, with a significant difference observed in the latter (1.9% vs. 6.6%, *p* = 0.01).

In terms of AKI staging, no significant differences were observed between the aciclovir and no-aciclovir groups in both the unmatched (*p* = 0.30) and matched cohorts (*p* = 0.07). The requirement for renal replacement therapy (RRT) was significantly lower in the aciclovir group compared to the no-aciclovir group in both the unmatched (0.8% vs. 20.6%, *p* < 0.001) and matched cohorts (0.8% vs. 25.1%, *p* < 0.001).

### 3.2. Multivariate Linear Regression Analysis

Multivariate linear regression analysis was conducted to further explore the risk factors for the primary outcome (see Table 3 and Table 4). Aciclovir treatment in the primary multivariate analysis was statistically significantly associated with a decrease in creatinine levels during ICU admission (beta-coefficient −14.5, 95% confidence interval (CI): −28.3–−0.68, *p* = 0.04). On the contrary, the cumulative dose of aciclovir was not found to be a risk factor for a change in creatinine levels (beta-coefficient: 0.001 (95% CI: −0.0005–0.002), *p* = 0.19). It was also found that RTT during ICU admission in the primary analysis is a statistically significant risk factor associated with a change in creatinine.

## 4. Discussion

In this retrospective study, we explored the potential nephrotoxic effect of aciclovir in a real-world ICU setting. Our results indicate that (TDM-driven) aciclovir therapy does not significantly impact renal function markers, suggesting that aciclovir treatment is generally safe in this critically ill population.

Interestingly, our data do not show a decline in renal function associated with aciclovir therapy, a finding that contrasts with some previous reports of nephrotoxicity [9,10,13]. This absence of renal dysfunction may be attributable to heightened clinical awareness and rigorous monitoring of renal function in ICU patients, which allows for timely adjustments in therapy to prevent toxicity. Clinicians’ vigilance in preventive management, including fluid supplementation and infusion duration and monitoring drug levels and kidney function, likely may have mitigated the development of nephrotoxic effects. Moreover, our data suggest that treatment with aciclovir might even enhance renal function during ICU stays. This observed improvement could potentially be attributed to the administration of fewer nephrotoxic co-medications within the aciclovir group or, alternatively, to the more extensive fluid supplementation provided to these patients relative to the control group. However, it should be noted that our study did not distinguish between prerenal, renal, and postrenal causes of kidney dysfunction, which may limit the ability to fully interpret these findings.

In the context of aciclovir-induced nephrotoxicity, certain risk factors for impaired renal function have been well documented in the product label of aciclovir. These include: pre-existing renal impairment, reduced fluid intake, and rapid infusion of aciclovir (bolus injections) [4]. The latter two suggest that nephrotoxicity is likely related to peak drug concentrations. To mitigate this risk, aciclovir is administered in a controlled one-hour infusion, with appropriate hydration protocols in place to protect renal function [21]. In our clinical setting, the drug is given as a short infusion over the course of one hour. Hydration protocols, crucial in this context, are especially emphasized in ICU patients, who are closely monitored for fluid balance. A fourth method to possibly mitigate adverse events of aciclovir could be by means of TDM. Additionally, the metabolite 9-carboxymethoxymethylguanine (CMMG), found in both plasma and cerebrospinal fluid, has been associated with the development of neurotoxicity [14,22]. TDM of both aciclovir and its metabolite may play a key role in preventing supratherapeutic drug exposures and minimizing the risks of adverse renal and neurological effects, though specific exposure–toxicity relationships for both substances are not yet well established [14]. Our cohort showed median aciclovir concentrations within exploratory therapeutic ranges, further supporting the safety profile of aciclovir when managed with appropriate precautionary measurements. However, our study was unable to assess the direct impact of TDM on clinical outcomes, underscoring the need for future research aimed at establishing therapeutic and toxic concentration thresholds for aciclovir and its metabolite.

Previous studies have identified several independent risk factors for aciclovir-induced nephrotoxicity, including age, cumulative drug dose, pre-existing renal dysfunction, and concurrent use of other nephrotoxic medications [13,15,17,23,24,25]. Older age, in particular, has been linked to reduced glomerular filtration rates and increased risk of aciclovir crystal deposition in the renal tubules, leading to nephrotoxicity. Moreover, older patients have an altered body composition compared to younger patients, and therefore the pharmacokinetic profile of hydrophilic drugs like aciclovir might be different, potentially resulting in higher drug levels [15]. Nonetheless, in our study, age did not emerge as a significant risk factor for increases in serum creatinine during aciclovir treatment. However, the predominance of patients over 60 years in our cohort may have masked any potential age-related impact on renal function.

The duration of aciclovir treatment has also been debated as a potential risk factor for nephrotoxicity since longer treatment courses and higher cumulative doses of aciclovir have been associated with an increased risk of nephrotoxicity [25]. However, our findings did not reveal an association between the cumulative dose of aciclovir and the development of nephrotoxicity. This reinforces the notion that, when monitored appropriately, aciclovir can be used safely even over extended treatment periods in critically ill patients. Although our study suggests that aciclovir is not an independent driver of nephrotoxicity, it remains crucial to closely monitor patients with pre-existing risk factors for renal impairment.

This study has several strengths and limitations. As one of the first to examine the relationship between aciclovir use and nephrotoxicity in a critically ill population, it provides novel insights into the renal safety of this antiviral therapy in ICU settings. We employed comprehensive medical records and robust statistical methodologies, including propensity score matching, to control for confounding variables. However, the retrospective design introduces the possibility of bias, and missing data, though limited, could affect the accuracy of our findings. Notably, our analysis relied solely on serum creatinine levels and use of RRT to define and stage acute kidney injury (AKI), despite the availability of other markers. Moreover, some biomarkers such as cystatin C might estimate the true renal function better in critically ill patients [26]. However, these markers are not readily available in every hospital and could complicate the comparison of study results altogether. Additionally, only a small portion of study participants had prior nephrotoxic complications (CKD in both groups <10%), which limited our ability to perform a meaningful subgroup analysis. Nonetheless, we collected multiple lab and clinical markers to detect nephrotoxicity, strengthening the reliability of our conclusions. The large sample size and consistency across analyses further support our findings. In addition, the use of average daily renal impairment values in our analysis provides a more nuanced and accurate estimate of renal function changes compared to studies that only consider peak values of laboratory renal markers during treatment.

These findings are particularly relevant for ICU settings where critically ill patients often require antiviral therapy. The observed safety of aciclovir, when administered with appropriate monitoring protocols, supports its use as a viable treatment option in this vulnerable population. By minimizing the risk of nephrotoxicity, TDM-driven aciclovir therapy may contribute to improved patient outcomes and lower complication rates associated with renal dysfunction, ultimately reducing ICU length of stay and healthcare costs.

The broader implications of these findings warrant further exploration. Future studies should aim to establish validated therapeutic and toxic concentration thresholds for aciclovir and its metabolite, CMMG, to optimize dosing strategies and minimize risks. Moreover, it is crucial to assess the generalizability of our findings to other patient populations, such as non-ICU settings or immunocompromised individuals receiving long-term antiviral therapy. Investigating the pharmacoeconomic impact of implementing TDM in clinical practice will also be vital to understanding its cost-effectiveness and feasibility on a larger scale. Elucidating the role of TDM in optimizing outcomes for patients at high risk of nephrotoxicity, including those with pre-existing renal impairment or polypharmacy, represents a critical area for future research. By addressing these gaps, future work can further enhance the safety profile of aciclovir therapy and inform best practices for its use in diverse clinical contexts.

All analyses, including propensity score matching, consistently demonstrated that aciclovir therapy did not compromise renal function in critically ill patients. This finding underscores the importance of vigilance in monitoring and preventative measures to mitigate nephrotoxic risks. However, caution is warranted when extrapolating our results to other patient populations or clinical indications, as pharmacokinetics may differ in other contexts.

## 5. Conclusions

In conclusion, our findings suggest that (TDM-driven) aciclovir treatment does not independently induce nephrotoxicity in well-hydrated ICU patients. Careful monitoring is necessary for those with pre-existing renal risk factors. Future research should focus on establishing validated therapeutic and toxic ranges for aciclovir to optimize dosing and minimize the risk of renal impairment in ICU settings. Additionally, studies investigating the added value and cost-effectiveness of TDM in this context and specific (nephrologically challenged) populations are warranted.

## Figures and Tables

**Figure 1 jcm-14-01409-f001:**
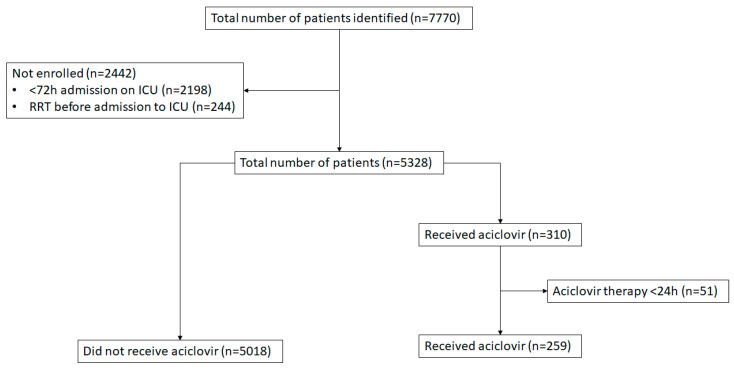
Flowchart of the selected patients.

**Table 1 jcm-14-01409-t001:** Baseline characteristics of patients in both the (propensity score) unmatched and matched cohorts. Bonferroni-corrected *p*-values beneath the significance level of 0.0026 are bolded. Values are presented as number (percentages) or as median [interquartile range] as appropriate.

	Unmatched Cohort			Matched Cohort		
	Aciclovir (n = 259)	No-Aciclovir (n = 5018)	*p*-Value	Aciclovir (n = 259)	No-Aciclovir (n = 259)	*p*-Value
**Demographic data**						
Age on admission, years	68 [60–74]	69 [60–76]	0.16	68 [60–74]	66 [54–75]	0.15
Gender, male	117 (45.2%)	3233 (64.4%)	0.22	117 (45.2%)	170 (65.6%)	0.58
BMI kg/m^2^	26.1 [23.8–29.4]	26.3 [23.7–30.1]	0.52	26.1 [23.8–29.4]	25.6 [22.6–29.8]	0.24
Obesity (yes)	55 (21.2%)	1305 (26.0%)	0.10	55 (21.2%)	63 (24.3%)	0.46
Immune deficient at baseline	65 (25.1%)	484 (9.6%)	**<0.001**	65 (25.1%)	38 (14.7%)	0.004
Diabetes	57 (22.0%)	1162 (23.2%)	0.72	57 (22.0%)	55 (21.2%)	0.92
CKD on admission	24 (9.2%)	478 (9.5%)	0.98	24 (9.2%)	17 (6.6%)	0.33
AKI on admission	51 (19.7%)	862 (17.2%)	0.34	51 (19.7%)	41 (15.8%)	0.30
Nephrotoxic agents at baseline			**<0.001**			0.074
0	208 (80.3%)	3218 (64.1%)		208 (80.3%)	196 (75.7%)	
1	50 (19.3%)	1589 (31.7%)		50 (19.3%)	56 (21.6%)	
2	1 (0.4%)	203 (4.0%)		1 (0.4%)	7 (2.7%)	
3	0	8 (0.2%)		0	0	
LOS ICU (days)	16.5 [9.7–30.0]	5.9 [4.0–10.5]	**<0.001**	16.5 [9.7–30.0]	11.1 [5.5–29.3]	**<0.001**
**Laboratory**						
Creatinine (µmol/L)	99.0 [67.5–172]	101 [73–157]	0.61	99.0 [67.5–172]	92.0 [65.0–140.5]	0.10
Calculated eGFR (mL/min)	56.8 [34.3–92.8]	59.7 [35.0–86.1]	0.49	56.8 [34.3–92.8]	65.8 [40.8–94.4]	0.07
Urea (mmol/L)	16.0 [9.7–24.9]	9.9 [6.8–15.9]	**<0.001**	16.0 [9.7–24.9]	9.65 [6.4–13.8]	**<0.001**
Diuresis on admission (mL/day)	1630 [1062–2642]	1645 [1050–2514]	0.51	1630 [1062–2642]	1675 [1102–2505]	0.89
**Severity scores**						
APACHE III	80 [67.5–100.5]	76 [60–95]	**<0.001**	80 [67.5–100.5]	86 [67.5–104.5]	0.44
APACHE IV probability	0.35 [0.23–0.61]	0.27 [0.11–0.52]	**<0.001**	0.35 [0.23–0.61]	0.37 [0.16–0.67]	0.81
SOFA score at baseline	7 [5–10]	8 [6–10]	0.06	7 [5–10]	7 [6–9]	0.58
Renal SOFA			0.96			0.23
0	146 (56.4%}	2771 (55.2%)		146 (56.4%}	154 (59.5%)	
1	44 (17.0%)	1019 (20.3%)		44 (17.0%)	55 (21.2%)	
2	29 (11.2%)	496 (9.9%)		29 (11.2%)	21 (8.1%)	
3	15 (5.8%)	224 (4.5%)		15 (5.8%)	10 (3.9%)	
4	25 (9.7%)	508 (10.1%)		25 (9.7%)	19 (7.3%)	
RIFLE score			0.02			0.67
0	152 (58.7%)	2420 (48.2%)		152 (58.7%)	141 (54.4%)	
1	22 (13.8%)	386 (7.7%)		22 (13.8%)	14 (5.4%)	
2	27 (10.4%)	911 (18.2%)		27 (10.4%)	50 (19.3%)	
3	20 (7.7%)	861 (17.2%)		20 (7.7%)	37 (14.3%)	
4	1 (0.4%)	66 (1.3%)		1 (0.4%)	4 (1.5%)	
5	37 (14.3%)	367 (7.3%)		37 (14.3%)	13 (5.0%)	
Mechanical ventilation on admission	219 (84.6%)	4117 (82.0%)	0.34	219 (84.6%)	214 (82.6%)	0.64
**Aciclovir dosing**						
Aciclovir therapy duration (days)	5.5 [3.2–9.0]	0	-	5.5 [3.2–9.0]	-	-
Cumulative aciclovir dose, mg	5500 [3000–8950]	0	-	5500 [3000–8950]	-	-

BMI: body mass index, SOFA: sequential organ failure assessment, RIFLE: Risk, Injury, Loss, Failure, End-stage renal disease, CKD: chronic kidney disease, AKI: acute kidney injury, LOS: length of stay, eGFR: estimated glomerular filtration rate.

**Table 2 jcm-14-01409-t002:** Univariate linear regression analyses for the primary and secondary objective(s) for both the (propensity score) unmatched and matched cohorts. Values are presented as number (percentages) or as median [interquartile range] as appropriate. Statistical significant outcomes are bolted.

	Unmatched Cohort			Matched Cohort		
	Aciclovir (n = 259)	No-Aciclovir (n = 5018)	*p*-Value	Aciclovir (n = 259)	No-Aciclovir (n = 259)	*p*-Value
**Primary outcome**						
Difference in creatinine (µmol/L)	−10.5 [−34.1–2.50]	−4.4 [−19.7–7.40]	0.09	−10.5 [−34.1–2.5]	−5.5 [−22.4–3.1]	0.22
**Secondary outcome**						
Difference in urea (mmol/L)	−0.49 [−3.9–3.5]	2.0 [−0.4–4.8]	**<0.001**	−0.49 [−3.9–3.5]	1.9 [−0.5–5.4]	**<0.001**
Difference in SOFA score	−1.3 [−3.0–0.1]	−1.9 [−3.2–0.8]	**<0.001**	−1.3 [−3.0–0.1]	−2.0 [−3.5–0.7]	**<0.001**
Difference in RIFLE score	0 [−0.3–0.1]	0 [−0.8–0.4]	0.71	0 [−0.3–0.1]	0 [−0.6–0.5]	0.44
Difference in eGFR (mL/min/1.73 m^2^)	6.3 [−1.3- 17.5]	2.5 [−3.9–9.7]	**<0.001**	6.3 [−1.3- 17.5]	3.31 [−2.6–14.1]	0.23
AKI, incidence	5 (1.9%)	220 (4.4%)	0.08	5 (1.9%)	17 (6.6%)	0.01
Stage AKI			0.30			0.07
1	2 (0.8%)	90 (1.8%)		2 (0.8%)	7 (2.7%)	
2	2 (0.8%)	87 (1.7%)		2 (0.8%)	5 (1.9%)	
3	1 (0.4%)	43 (0.9%)		1 (0.4%)	5 (1.9%)	
RRT, incidence	2 (0.8%)	1034 (20.6%)	**<0.001**	2 (0.8%)	65 (25.1%)	**<0.001**

SOFA: sequential organ failure assessment, RIFLE: Risk, Injury, Loss, Failure, End-stage renal disease, AKI: acute kidney injury, RRT: renal replacement therapy, eGFR: estimated glomerular filtration rate.

**Table 3 jcm-14-01409-t003:** Multivariate linear regression analysis of the (propensity score) matched cohort. Statistically significant outcomes are bolded.

	Beta-Coefficient	Std. Error	T-Statistic	*p*-Value	95% CI Lower (b-coef)	95% CI Upper (b-coef)
Aciclovir therapy, yes	−14.5	7.03	−2.06	**0.04**	−28.3	−0.68
Propensity score	−15.7	17.4	−0.90	0.37	−50.0	18.5
RRT, yes	−27.1	10.6	−2.56	**0.01**	−47.9	−6.27

RRT: renal replacement therapy, std. error: standard error, b-coef: beta-coefficient.

**Table 4 jcm-14-01409-t004:** Multivariate linear regression analysis of the (propensity score) matched cohort. Statistically significant outcomes are bolded.

	Beta-Coefficient	Std. Error	T-Statistic	*p*-Value	95% CI Lower (b-coef)	95% CI Upper (b-coef)
Cumulative aciclovir dose, mg	0.001	0.001	1.31	0.19	−5.0 × 10^−4^	0.002
Propensity score	−15.7	17.4	−0.90	0.54	−61.3	32.3
RRT, yes	−27.1	10.6	−2.56	0.89	−99.2	114

RRT: renal replacement therapy.

## Data Availability

Data are available on request due to legal and privacy restrictions.

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
