# Peer review of "Evaluation of Aciclovir-Induced Nephrotoxicity in Critically Ill Patients: A Propensity-Matched Cohort Study"

_jcm, 2025, doi:10.3390/jcm14051409_

Round 1

Reviewer 1 Report

Comments and Suggestions for Authors

This study employs a propensity-matched cohort design to address a clinically relevant concern that acyclovir-induced nephrotoxicity in critically ill patients, a population vulnerable to drug-induced renal injury. It suggests that under TDM and controlled administration protocols, aciclovir may not independently lead to nephrotoxicity and might even improve renal function due to associated care measures. These findings are promising, but the retrospective design, traditional renal markers, and unaddressed confounders necessitate further investigation.

Here are some suggestions for improving the manuscript.

1.     Describe the significance of the study in the abstract.

2.     Replace "p < 0.001" with the exact p-values throughout the text and tables. Ensure consistency in reporting statistical significance across all results.

3.     Please provide the value of p instead of p<0.001 in the context and tables.

4.     Consider discussing the broader implications of your findings.

5.     Line 105: Expand the methodological details of the UHPLC-MS/MS, such as instrument model, the volume of serum and acetonitrile used for sample preparation, the concentration of acyclovir-d4, the brand, length, and size of chromatography column, the elution gradient (e.g., timing, solvent ratios), the MS/MS parameters (e.g., ionization mode, mass to charge ratio, transitions), etc.

6.     Replace “analysi with “analysis” in line 181. Perform a thorough proofreading of the manuscript to catch similar typographical errors.

Author Response

This study employs a propensity-matched cohort design to address a clinically relevant concern that acyclovir-induced nephrotoxicity in critically ill patients, a population vulnerable to drug-induced renal injury. It suggests that under TDM and controlled administration protocols, aciclovir may not independently lead to nephrotoxicity and might even improve renal function due to associated care measures. These findings are promising, but the retrospective design, traditional renal markers, and unaddressed confounders necessitate further investigation.

Here are some suggestions for improving the manuscript.

Comment 1: Describe the significance of the study in the abstract.

Response 1:  We thank the reviewer for pointing this out. We have revised the abstract to explicitly describe the significance of the study. The adjusted abstract highlights the clinical relevance and the potential impact of the findings on the safe use of aciclovir in critically ill ICU patients, emphasizing the role of therapeutic drug monitoring (TDM) in mitigating nephrotoxicity risks. This addition addresses the broader implications of the study and its contribution to the field.

We added the following to the conclusion part of the abstract: The results are significant as they provide evidence supporting the safe use of aciclovir in vulnerable/critically-ill patients.

Comment 2: Replace "p < 0.001" with the exact p-values throughout the text and tables. Ensure consistency in reporting statistical significance across all results.

Response 2: We strongly agree with the reviewer that consistancy in reporting p-values throughout the manuscript is important. However the p-values we found are often <<< 0.001 (e.g. 2.2*10-16). Since we consider the added value of reporting the exact numbers is rather small, we prefer to keep a threshold on the reported number for the p-value to improve the radability of the manuscript. However, we have meticously checked the consistancy of the p-values throughout the manuscript and made changes in accordance with the suggestion of the reviewer.

Comment 3: Replace "p < 0.001" with the exact p-values throughout the text and tables. Ensure consistency in reporting statistical significance across all results.

Response 3: We once again thank the reviewer for this comment, for the changes and response we refer the ‘Response 2’.

Comment 4:  Consider discussing the broader implications of your findings.

Response 4: Thank you for your valuable suggestion. We have revised the discussion to include the broader implications of our findings for clinical practice and future research. Specifically, we have highlighted the relevance of our results for optimizing aciclovir therapy in ICU settings, the potential economic benefits of TDM, and the need for further research into its application in various patient populations.

Comment 5:     Line 105: Expand the methodological details of the UHPLC-MS/MS, such as instrument model, the volume of serum and acetonitrile used for sample preparation, the concentration of acyclovir-d4, the brand, length, and size of chromatography column, the elution gradient (e.g., timing, solvent ratios), the MS/MS parameters (e.g., ionization mode, mass to charge ratio, transitions), etc.

Response 5: We agree with the reviewer it is important for the reproducibility of the study to expand the methodology of the bioanalysis of aciclovir. We therefore added the suggested details into the method section of the manuscript.

Comment 6:    Replace “analysi” with “analysis” in line 181. Perform a thorough proofreading of the manuscript to catch similar typographical errors.

Response 6: We thank the reviewer for point out the typographical errors. Additionally, a thorough proofreading of the manuscript has been performed to identify and correct similar typographical errors throughout the text. We appreciate your attention to detail and believe this revision further enhances the clarity and readability of the manuscript.

Reviewer 2 Report

Comments and Suggestions for Authors

This study monitors the effect of acyclovir treatment in critically ill patients with special reference to the nephrotoxic effect. It describes a comprehensive approach to assess differences between the characteristics of patients divided into two groups, based on whether patients admitted to intensive care were treated with acyclovir or not. By using the appropriate statistical tests, especially the propensity score matching method, the confusion between the mentioned groups of patients was effectively reduced, making the comparison more reliable. The study found a lower incidence of acute kidney injury (AKI) and the need for renal replacement therapy (RRT) in the acyclovir group, supporting the hypothesis that acyclovir may have a protective effect in renal function in critically ill patients. It is encouraging to note that acyclovir, when administered with appropriate protocols, does not significantly affect renal function, especially given concerns about its nephrotoxicity. However, while acyclovir appears to show some potential benefits in terms of renal protection and reduced need for renal replacement therapy, further studies involving larger sample sizes or additional control measures, as well as those evaluating the direct effect of therapeutic drug monitoring on clinical outcomes, are needed to confirm these findings and elucidate the precise mechanisms behind these outcomes. In conclusion, this study adds important evidence supporting the safe use of acyclovir in intensive care units, particularly in combination with preventive measures including rigorous monitoring, hydration, and timely adjustment of therapy, especially for those with pre-existing renal risk factors.

Minor issues:

- there are some minor typographical issues, such as "multivarate" instead of "multivariate", "bèta-coefficient" instead of "beta“, „analysis“ instead od „analysi“, incorrectly placed brackets „(CKD-EPI) [19]“ instead of „(CKD-EPI [19])“ e

- in the introducton section, the authors should list other toxic consequences of acyclovir in addition to nephrotoxicity in order to present the effect of acyclovir accumulation more comprehensively

- OLVG is an abbreviation that needs to be explained when it first appears in the text

- since the references are not written according to the instructions for authors, appropriate corrections should be made.

The are also some proposed corrections are listed in the attachment.

Author Response

This study monitors the effect of acyclovir treatment in critically ill patients with special reference to the nephrotoxic effect. It describes a comprehensive approach to assess differences between the characteristics of patients divided into two groups, based on whether patients admitted to intensive care were treated with acyclovir or not. By using the appropriate statistical tests, especially the propensity score matching method, the confusion between the mentioned groups of patients was effectively reduced, making the comparison more reliable. The study found a lower incidence of acute kidney injury (AKI) and the need for renal replacement therapy (RRT) in the acyclovir group, supporting the hypothesis that acyclovir may have a protective effect in renal function in critically ill patients. It is encouraging to note that acyclovir, when administered with appropriate protocols, does not significantly affect renal function, especially given concerns about its nephrotoxicity. However, while acyclovir appears to show some potential benefits in terms of renal protection and reduced need for renal replacement therapy, further studies involving larger sample sizes or additional control measures, as well as those evaluating the direct effect of therapeutic drug monitoring on clinical outcomes, are needed to confirm these findings and elucidate the precise mechanisms behind these outcomes. In conclusion, this study adds important evidence supporting the safe use of acyclovir in intensive care units, particularly in combination with preventive measures including rigorous monitoring, hydration, and timely adjustment of therapy, especially for those with pre-existing renal risk factors.

Minor issues:

Comment 1: there are some minor typographical issues, such as "multivarate" instead of "multivariate", "bèta-coefficient" instead of "beta“, „analysis“ instead od „analysi“, incorrectly placed brackets „(CKD-EPI) [19]“ instead of „(CKD-EPI [19])“ e

Response 1: We thank the reviewer for pointing out these typographical issues, we corrected them throughout the manuscript. Additionally, a thorough proofreading of the manuscript has been performed to identify and correct similar typographical errors throughout the text. We appreciate your attention to detail and believe this revision further enhances the clarity and readability of the manuscript.

Comment 2: in the introducton section, the authors should list other toxic consequences of acyclovir in addition to nephrotoxicity in order to present the effect of acyclovir accumulation more comprehensively

Response 2: The reviewer indeed is correct that aciclovir treatment can lead to other toxicities, we have included these additional adverse effects (neurotoxicity, gastrointestinal toxicity and hematological toxicity) to the introduction. This provides a more holistic view of aciclovir's potential toxic consequences.

Comment 3: OLVG is an abbreviation that needs to be explained when it first appears in the text

Response 3: We thank the author for this critical note. OLVG used to be the abbreviation of ‘Onze Lieve Vrouw Gasthuis’, however in 2015 the Onze Lieve Vrouw Gasthuis and the Sint Lucas Andreas hospital merged and adopted the abbreviation as its new name.  

Comment 4: since the references are not written according to the instructions for authors, appropriate corrections should be made.

Response 4: The reviewer is right for pointing this out. The references are reformated based on the instruction for authors of both the J Clin Med as the MPDI.

Comment 5: The are also some proposed corrections are listed in the attachment.

Response 5: We once again thank the reviewer, we included al the suggested corrections in the manuscript.

Reviewer 3 Report

Comments and Suggestions for Authors

To the authors, 

Thank you for the opportunity to review this manuscript. This study examined the safety of acyclovir in ICU patients and demonstrated that acyclovir use did not contribute to subsequent renal dysfunction. Therefore, this is an important and clinically relevant topic. However, the following issues need to be addressed: 

<Specific comments>

1. Propensity score matching was used to adjust for patient background. However, the population requiring acyclovir is likely to be a highly specific subgroup (i.e., patients requiring antiviral therapy). The control group consisted of ICU patients with diverse backgrounds, which may have introduced differences in patient characteristics when compared to the acyclovir group. The authors should provide details regarding the reasons for ICU admission to clarify the comparability of the groups. 

2. The toxicity of acyclovir may depend on its dose relative to body weight. An analysis of the acyclovir dosage per body weight should be included to strengthen the conclusions of the study.

Author Response

Comment 1

Propensity score matching was used to adjust for patient background. However, the population requiring aciclovir is likely to be a highly specific subgroup (i.e., patients requiring antiviral therapy). The control group consisted of ICU patients with diverse backgrounds, which may have introduced differences in patient characteristics when compared to the aciclovir group. The authors should provide details regarding the reasons for ICU admission to clarify the comparability of the groups.

Response 1

We thank the reviewer for raising this important point. We have thoroughly examined the reasons for ICU admission in both the aciclovir and the non-aciclovir groups. Since aciclovir is administered both as treatment and as prophylaxis in the ICU, the indications for ICU admission are heterogeneous. In the aciclovir group (n=259), there were 68 different admission indications, while in the non-aciclovir group, there were 76. However, the top five admission diagnoses were similar in both groups: pneumonia, sepsis, cardiac arrest, coronary artery bypass graft (CABG) surgery, and congestive heart failure.

Moreover, ICU patients typically present with multiple comorbidities, making patient characteristics inherently complex and diverse. This heterogeneity complicates direct comparisons. Nevertheless, the main objective of our study is to assess the impact of aciclovir on renal function. To address potential confounding factors, we included comorbidities that influence renal function in our analysis.

Comment 2

The toxicity of aciclovir may depend on its dose relative to body weight. An analysis of the aciclovir dosage per body weight should be included to strengthen the conclusions of the study.

We appreciate the reviewer’s suggestion and agree that analyzing aciclovir dosage per kilogram of body weight could provide additional insights. However, we argue that aciclovir is a hydrophilic compound with a volume of distribution largely limited to systemic circulation and extracellular water. Consequently, drug exposure may actually be higher in patients with increased body mass (especially in cases of obesity), which raises the question of whether the current weight-based dosing strategy sufficiently reduces inter-individual variability in drug exposure. While this is an interesting consideration, it falls beyond the scope of this manuscript.

Additionally, since therapeutic drug monitoring (TDM) is performed, patients with higher aciclovir concentrations are typically dose-adjusted accordingly, which further complicates the interpretation of a direct mg/kg analysis. Nevertheless, we performed the suggested analysis and included the data for the reviewer’s consideration (see table 1 below). Our findings indicate that aciclovir dosage per body weight is not statistically significantly associated with a decline in renal function.

Table 1. Multivariate linear regression analysis of the (propensity score) matched cohort. Statistical significant outcomes are bolted.

Beta-coefficient

Std. Error

T- Statistic

p-value

95% CI Lower

(b-coef)

95% CI Upper

(b-coef)

Aciclovir mg/kg dose

0.001

0.001

1.31

0.19

-0.0005

0.002

Propensity score

-14.5

23.8

-0.61

0.54

-61.3

32.3

RRT, yes

7.18

54.0

0.13

0.89

-99.2

114

Reviewer 4 Report

Comments and Suggestions for Authors

Dear Authors,

Respectfully, I have shared my comments and suggestions, which I hope will contribute to enhancing the clarity, coherence, and scientific rigor of your manuscript. Below, I outline key areas for improvement and propose practical recommendations.

Title

Although the current title is methodologically correct, I suggest adopting a more neutral language regarding nephrotoxicity. Example: Evaluation of aciclovir-induced nephrotoxicity in critically ill patients: A propensity-matched cohort study.

Statistical analysis

Comment: There is no explicit mention of a test for normality verification.

Comment: I recommend including the analysis of Standardized Mean Differences (SMDs) and Love plots in the supplementary material. This approach is a practical solution to demonstrate the robustness of the Propensity Score Matching (PSM) without overloading the main text.

Primary Endpoint

Comment: Why was ΔCreatinine, instead of more sensitive markers such as ΔeGFR, chosen as the primary endpoint to evaluate renal function in this study?

Results

Line 181: Insert the results description before Table 2 and cite it in the text (Table 2).

Line 213: Insert the results description before Table 3 and cite it in the text (Table

Line 216: Insert the results description before Table 4 and cite it in the text (Table 4).

Discussion

Comment: Justify why ΔCreatinine, which is less sensitive than ΔeGFR, was chosen as the primary endpoint. This would clarify whether the choice was based on methodological limitations or clinical considerations.

Comment: Incorporate a more detailed analysis of eGFR, a more sensitive marker adjusted for age, sex, and muscle mass, and justify why it was considered a secondary endpoint.

Comment: Noting that subgroups like CKD (6.6%) were insufficient for robust analyses is a valid comment.

Limitation

Comment: I suggest considering acknowledging, in the limitations section, the absence of interaction terms in the multivariate model, such as Aciclovir and CKD or Aciclovir and Nephrotoxics. Including this observation would enhance methodological transparency and highlight the importance of exploring potential combined effects or effect modifiers in future studies.

Conclusion

Comment: Highlight that the results may be more relevant for ICUs with similar hydration and TDM protocols, limiting their generalizability to other contexts.

Comment: Emphasize that small subgroups, such as patients with CKD, may not have been adequately represented.

Author Response

Dear Authors,

Respectfully, I have shared my comments and suggestions, which I hope will contribute to enhancing the clarity, coherence, and scientific rigor of your manuscript. Below, I outline key areas for improvement and propose practical recommendations.

Comment 1: Title

Although the current title is methodologically correct, I suggest adopting a more neutral language regarding nephrotoxicity. Example: Evaluation of aciclovir-induced nephrotoxicity in critically ill patients: A propensity-matched cohort study.

Response 1: We agree and changed the title to the suggested one.

Statistical analysis

Comment 2: There is no explicit mention of a test for normality verification.

Response 2: We thank the author for pointing this out, indeed we did not mention the normality testing. We added to the manuscript that we used the Shapiro-Wilk test for testing whether the data is normally distributed.

Comment 3: I recommend including the analysis of Standardized Mean Differences (SMDs) and Love plots in the supplementary material. This approach is a practical solution to demonstrate the robustness of the Propensity Score Matching (PSM) without overloading the main text.

Response 3: We agree that it is important to show the robustness of the methodology. However, we believe that detailed statistic descriptions are outside of the scope of the Journal of Clinical Medicine. Nonetheless, we included the loveplot as supplementary material.

Primary Endpoint

Comment 4: Why was ΔCreatinine, instead of more sensitive markers such as ΔeGFR, chosen as the primary endpoint to evaluate renal function in this study?

Response 4: The reviewer raises a valid question. In our study, we used ΔCreatinine rather than ΔeGFR as the primary endpoint to evaluate renal function for several reasons. First, creatinine is a more robust and direct parameter to assess renal function, avoiding potential inaccuracies introduced by eGFR algorithms. Notably, eGFR formulas, such as CKD-EPI, are validated in specific populations—for instance, CKD-EPI is validated primarily in individuals with an eGFR <90 mL/min. Applying such algorithms to our ICU population, which includes patients with varying and often transient renal function, could lead to inaccurate estimations for approximately 25% of the study population.

Furthermore, in critically ill ICU patients, factors like fluid shifts, altered muscle mass, and medication effects can compromise the accuracy of eGFR calculations, making ΔCreatinine a more reliable marker in this setting. ΔCreatinine also aligns with methods used in prior nephrotoxicity studies involving ICU patients, ensuring consistency with established research practices. Finally, retrospective studies often lack comprehensive demographic data required for eGFR calculation, but serum creatinine levels are universally available and unaffected by these limitations.

The reviewer is however correct that creatinine is not an ideal biomarker, therefore we added to the discussion the limitation of the use of creatinine in critically ill patients.

We hope this clarification addresses the reviewer’s concerns and demonstrates the robustness of our methodological approach.

Results

Comment 5:

Line 181: Insert the results description before Table 2 and cite it in the text (Table 2).

Line 213: Insert the results description before Table 3 and cite it in the text (Table

Line 216: Insert the results description before Table 4 and cite it in the text (Table 4).

Response 5: We thank the reviewer for the suggestion to move the text in the result section above the tables. We agree with this suggestion and implemented these in the revised manuscript.

Discussion

Comment 6a: Justify why ΔCreatinine, which is less sensitive than ΔeGFR, was chosen as the primary endpoint. This would clarify whether the choice was based on methodological limitations or clinical considerations.

Comment 6b: Incorporate a more detailed analysis of eGFR, a more sensitive marker adjusted for age, sex, and muscle mass, and justify why it was considered a secondary endpoint.

Response 6: We agree with the reviewer the choice in primary endpoint could be more elaborately discussed. We feel that the explanation could be better mentioned in the method section. Therefore we added: ‘Creatinine, rather than eGFR, was chosen as the primary endpoint due to its robustness, aligning with methods used in prior nephrotoxicity studies involving ICU patients, ensuring consistency with established research practices, see also the response given for comment 4.

Comment 7: Noting that subgroups like CKD (6.6%) were insufficient for robust analyses is a valid comment.

Response 7: The reviewer raises a valid point, we included this in the discussion section of the manuscript

Limitation

Comment 8: I suggest considering acknowledging, in the limitations section, the absence of interaction terms in the multivariate model, such as Aciclovir and CKD or Aciclovir and Nephrotoxics. Including this observation would enhance methodological transparency and highlight the importance of exploring potential combined effects or effect modifiers in future studies.

Response 8: We thank the reviewer for this critical note. We would like to highlight that we included different interactions (of both dependable and independable covariate) terms in the multivariate analysis. As stated in the manuscript only RRT was a statistical significant covariate. The reviewer is (based upon the earlier remarks) right that there is a small group of patients suffering CKD. Therefore, it would indeed be valuable to perform an analysis in a patient cohort with prior nephrological complications. We added this to the discussion of the manuscript.

Conclusion

Comment 9: Highlight that the results may be more relevant for ICUs with similar hydration and TDM protocols, limiting their generalizability to other contexts.

Response 9: We highlighted that adequate hydration is a necessity (this is conform the drug label).

Comment 10: Emphasize that small subgroups, such as patients with CKD, may not have been adequately represented.

Response 10: We added that patients with CKD have been excluded from this study to the conclusion section of the manuscript

Round 2

Reviewer 3 Report

Comments and Suggestions for Authors

Thank you for the opportunity to review the revised manuscript. It responded sincerely to the comments and suitable.